# Knowledge, attitudes and experiences of self-harm and suicide in low-income and middle-income countries: protocol for a systematic review

Rebecca McPhillips [ID],[1] Sadia Nafees [ID],[2] Anam Elahi,[1] Saqba Batool,[1] Murali Krishna,[3] Anne Krayer [ID],[3] Peter Huxley,[3] Nasim Chaudhry,[4] Catherine Robinson[5]

[1]Social Care and Society, The University of Manchester, Manchester, UK
[2]North Wales Centre for Primary Care Research, Bangor University, Bangor, UK
[3]Centre for Mental Health and Society, School of Health Sciences, Bangor University, Bangor, UK
[4]Research and Development, Pakistan Institute of Living and Learning, Karachi, Pakistan
[5]The University of Manchester, Manchester, UK

**Correspondence to**
Dr Rebecca McPhillips;
Rebecca.McPhillips@manchester.ac.uk

## ABSTRACT

**Introduction** Over 800 000 people die due to suicide each year and suicide presents a huge psychological, economic and social burden for individuals, communities and countries as a whole. Low-income and middle-income countries (LMICs) are disproportionately affected by suicide. The strongest risk factor for suicide is a previous suicide attempt, and other types of self-harm have been found to be robust predictors of suicidal behaviour. An approach that brings together multiple sectors, including education, labour, business, law, politics and the media is crucial to tackling suicide and self-harm. The WHO highlights that evaluations of the knowledge and attitudes that priority groups, not only healthcare staff, have of mental health and suicidal behaviour are key to suicide prevention strategies. The aim of this systematic review is to examine the knowledge, attitudes and experiences different stakeholders in LMICs have of self-harm and suicide.

**Methods and analysis** MEDLINE, Embase, PsycINFO, CINAHL, BNI, Social Sciences and Cochrane Library will be searched. Reviewers working independently of each other will screen search results, select studies for inclusion, extract and check extracted data, and rate the quality of the studies using the Strengthening the Reporting of Observational studies in Epidemiology and Critical Appraisals Skills Programme checklists. In anticipation of heterogeneity, a narrative synthesis of quantitative studies will be provided and metaethnography will be used to synthesise qualitative studies.

**Ethics and dissemination** Ethical approval is not required. A report will be provided for the funding body, and the systematic review will be submitted for publication in a high-impact, peer-reviewed, open access journal. Results will also be disseminated at conferences, seminars, congresses and symposia, and to relevant stakeholders.

**PROSPERO registration number** CRD42019135323.

## Strengths and limitations of this study

► A strength of this systematic review protocol is that it has been written according to the Preferred Reporting Items for Systematic review and Meta-Analysis (PRISMA-P) 2015 checklist.
► A strength of the review is that it will conform to the PRISMA statement and to the Cochrane systematic review literature guidelines when results are reported.
► A strength of this review is that both quantitative and qualitative evidence will be assessed.
► A limitation of the review is the inclusion of peer-reviewed studies only; however, language restrictions will not be applied.
► As it is likely that the quantitative studies included in the review will be heterogeneous, a limitation will be the lack of meta-analysis.

## INTRODUCTION

The World Health Organization (WHO) estimates that over 800 000 people die due to suicide each year; 1 person every 40 s.[1] Suicide disproportionately affects low-income and middle-income countries (LMICs). In 2014, the WHO reported that 75.5% of suicides globally occur in LMICs, and in South East Asia, suicide is the leading cause of death in 15–29 year olds.[1] However, the under-reporting and misclassification of suicide as a cause of death in LMICs mean that suicide rates are likely higher than reported.[1 2] Every suicide death is a tragedy for families, friends and communities and suicide presents huge psychological, economic and social burden for individuals, communities and countries as a whole.[1] Reducing suicide is a key indicator for the United Nations' sustainable development goal to ensure healthy lives and promote well-being at all ages globally.[3] However, much of the published literature on suicide relates to high-income countries (HICs), and to effect change a better understanding of suicide within the cultural, political and socioeconomic context of LMICs is needed. Patient profiles, suicide rates, aetiology and

methods differ between LMICs and HICs.[4] For example, research to date indicates that the ratio of women to men who die by suicide in LMICs is much lower than in HICs.[5] Furthermore, while marriage is considered to be a protective factor for women in HICs, it is less so for women in some LMICs, and self-immolation and the consumption of pesticides are far more common methods in LMICs than in HICs.[6–9]

The strongest risk factor for suicide is a previous suicide attempt, and the WHO suggests that for each adult who dies from suicide, there may be 20 others attempting suicide.[1] Harm arising from suicidal behaviour, suicide attempts and suicide are types of self-harm that are often differentiated from non-suicidal self-injury (NSSI) in terms of intent, frequency, methods, lethality and cognitions.[10] The motivation for suicidal behaviours is often to remove suffering and the intent of suicidal behaviours is to end one's life, whereas the intent of NSSI is not. NSSI behaviours are more frequent than suicide and suicide attempts, with individuals using more varying and less lethal methods, and it is suggested that the cognitions related to NSSI concern temporary relief while those related to suicidal behaviour concern permanent relief.[10–13] Similar to the literature on suicide, much of that concerning NSSI is focused on HICs,[14–16] where NSSI has been found to be a robust predictor of suicidal behaviour, with this link remaining after controlling for age, gender and ethnicity.[12 17 18] A systematic review of the limited empirical research on self-harm, including suicidal self-harm and NSSI, in LMICs found that the prevalence of NSSI and suicide attempts in LMICs was comparable to HICs, that the most common methods of NSSI in LMICs were hitting, cutting, wound picking and biting, and these findings were similar to evidence from HICs.[16] Risk factors identified for suicidal self-harm and NSSI in LMICs were often related to family, for example, family conflict, divorced parents and childhood abuse; and protective factors were high family functioning and understanding parents, which were attributed to greater reliance on family in LMICs compared with many Western HICs.[16]

Suicide and self-harm in both LMICs and HICs are the result of complex interactions between genetic, psychological, biological, cultural, sociodemographic and social factors.[1 19 20] Although the healthcare sector clearly has a vital role to play in tackling suicide and self-harm in LMICs, an approach that brings together multiple sectors, including education, labour, business, law, politics and the media is crucial.[1 21] The knowledge, attitudes and experiences that stakeholders from various sectors have of suicide and self-harm are likely to influence suicide and self-harm prevention and intervention strategies. A recent review by the WHO[21] highlights that evaluations of the knowledge and attitudes that priority groups, for example, policy makers and community groups, not only healthcare staff, have of mental health and suicidal behaviour are key to the collection of high quality surveillance data and prevention strategies. Reviews to date have focused on the knowledge, attitudes and experiences that healthcare professionals have towards self-harm and suicide.[22–25] The aim of this systematic review is to examine the knowledge, attitudes and experiences of self-harm and suicide of various stakeholders. Therefore, in addition to stakeholders from the healthcare sector, other stakeholders who will be included in this review are people who have experienced self-harm and/or have attempted suicide themselves, and their relatives, friends and co-workers, and stakeholders from the social, healthcare, government and criminal justice sectors. We are interested in exploring the range of publications on the broad spectrum of knowledge, attitudes and experiences that these various stakeholders may have concerning suicide and self-harm, including, for example, knowledge stakeholders may have on prevalence and risk and protective factors for suicide and self-harm, stigmatising or empathetic attitudes towards those who self-harm, and experiences such as providing or receiving medical treatment for self-harm. This systematic review is being undertaken as part of the South Asia Self Harm Initiative (SASHI) project, which aims to help to find effective responses to self-harm and suicide in South Asia by building capability and capacity in research infrastructure and expertise in the region. Findings from this systematic review will be used to inform the development of a survey on knowledge, attitudes and well-being in South Asia. Thus, we are particularly interested in studies conducted in South Asia and countries with comparable healthcare systems or cultural backgrounds.

### Research question

The Setting, Perspective, phenomena of Interest, Comparison, Evaluation framework was used to generate the research question that will be addressed by this systematic review:[26]

► What are stakeholders' knowledge, attitudes and experiences of self-harm and suicide in LMICs?

### METHODS AND ANALYSIS

This protocol conforms to the Preferred Reporting Items for Systematic Reviews and Meta-Analyses Protocols (PRISMA-P) checklist (online supplemental file 1).[27] We will conform to the PRISMA statement and to the Cochrane systematic review literature guidelines when reporting the results.[28 29] This systematic review has been registered on PROSPERO.[30]

### Search strategy

A Betsi Cadwaladr University Health Board (BCUHB) librarian with expertise in systematic reviews has assisted the authors in the development of the search strategy (online supplemental appendix 1). We will search MEDLINE, Embase, PsycINFO, CINAHL, BNI, Social Sciences and Cochrane Library. We will not apply any language restrictions to the search criteria. EndNote and Microsoft Word will be used to manage initial search results, screening and data throughout the review. We

will update the searches prior to publication to ensure the latest papers are included. Reference lists from included studies and any identified systematic or literature reviews will also be searched by hand. Study authors will be contacted in instances when it has not been possible to retrieve full-text articles and when clarification regarding inclusion criteria, for example, participant age, is required.

## Study selection criteria

Inclusion criteria are empirical studies conducted in LMICs, as defined by the Organisation for Economic Co-operation and Development,[31] irrespective of the study design, whose focus is on the knowledge, attitudes or experiences of stakeholders towards self-harm and/or suicide, where participants are aged 16 years and above. Studies that include stakeholders' knowledge, attitudes and experiences of suicide and self-harm related to those under 16 will be included. Stakeholders are people who have experienced self-harm and/or have attempted suicide themselves; relatives, friends, co-workers and healthcare workers of those who have self-harmed, attempted or completed suicide; and people in the social, healthcare, government and criminal justice sectors. Exclusion criteria are studies conducted in HICs and studies whose participants are not aged 16 years and above. Studies whose main focus is on the prevalence and/or predictors of self-harm and/or suicide, relationships between state and/or trait characteristics and self-harm and/or suicide, euthanasia, terrorism or epidemiology will also be excluded. Systematic and literature reviews will be consulted for relevant references, but will not be included in the review. Opinion pieces, editorials, book reviews, and conference and poster abstracts will not be included in the review.

The selection of studies for inclusion will adhere to the Cochrane guidelines and the process of selection of eligible studies will be illustrated via a PRISMA diagram.[29] Following deduplication of search results in EndNote, the following screening process will be undertaken in order to select studies for inclusion in the systematic review:

1. Titles and abstracts will be read by two reviewers independently, and relevance and fit with the inclusion criteria will be assessed. Those of no obvious relevance will be excluded and any disagreements will be resolved with a third reviewer (and the wider expert group if necessary).
2. Full-text articles of remaining studies will be retrieved and read by two reviewers independently to assess their suitability for inclusion in the final review, disagreements will be resolved by discussion with a third reviewer (and the wider expert group if necessary). Both reviewers will populate a piloted pro forma for each full-text paper read (online supplemental appendix 2).

## Data extraction

Data will be extracted from selected studies by one reviewer, and a second reviewer will check for accuracy.

Extracted data will be recorded on a piloted pro forma (online supplemental appendix 2), and will reflect the inclusion criteria and the designated aims of the review derived from the article as a whole. Discrepancies will be resolved through discussion (with the wider expert group if necessary). Additional data will be requested from study authors when necessary. Data extraction of qualitative studies (and for qualitative components in studies with mixed methods) will adhere to the same methods and will be reviewed independently.

## Outcomes

Outcomes of interest include:

► The identification of relevant information on stakeholders' knowledge, attitudes and experiences of self-harm and suicide, particularly in South Asia and in countries with comparable healthcare systems and cultural backgrounds.
► The quantitative methods and measures that have been used to investigate stakeholders' attitudes towards and knowledge about self-harm and suicide and their psychometric properties.
► The qualitative methods that have been used to investigate stakeholders' attitudes towards knowledge about, and experiences of self-harm and suicide.

The identified outcomes will inform the development of a survey on knowledge, attitudes and well-being in South Asia as part of the SASHI project.

## Quality assessment

All eligible studies will be subject to quality appraisal. The quality of included quantitative studies will be appraised using the Strengthening the Reporting of Observational studies in Epidemiology (STROBE) checklist.[32] The STROBE Statement consists of a checklist of 22 items, which relate to the title, abstract, introduction, methods, results and discussion sections of articles. Eighteen items are common to cohort studies, case–control studies and cross-sectional studies, and four are specific to each of the three study designs. The quality of included qualitative studies will be appraised using the Critical Appraisals Skills Programme (CASP) checklist.[33] The 10-item CASP tool was considered to be the most suitable tool to consider the quality parameters of qualitative work, and is a well-validated and accepted tool.[28] Both the STROBE and CASP checklists will be applied independently by two reviewers and any disagreements will be resolved with a third reviewer (and the wider expert group if necessary).

Studies will not be excluded on the basis of poor quality alone, rather all studies that meet the inclusion criteria will be included in the review. This low threshold for inclusion will be applied so that the review can benefit from researcher insight and theoretical as well as empirical contributions. The relative quality of included studies will be critically considered and discussed in the review.

## Descriptive analysis and data synthesis

We anticipate that the quantitative studies included in the review will be heterogeneous and this will prevent meta-analysis. We will provide a narrative synthesis of quantitative studies, structured around population characteristics and the geographical region of studies. We will provide summaries of the quantitative methods and measures used to investigate stakeholders' attitudes towards and knowledge about self-harm and suicide and their psychometric properties.

Metaethnography will be used to synthesise qualitative studies.[34] Initially, reciprocal translation will be performed by comparing the concepts presented in different studies. A chronological approach will be taken to reciprocal translation; studies will be arranged chronologically, concepts from papers one and two will be compared, and the synthesis of papers one and two will then be compared with paper three, and so forth, as is described elsewhere.[35] When contradictions between studies are identified, we will perform refutational synthesis by exploring and explaining these. A 'lines-of-argument' synthesis, that links and explains concepts presented by different studies, will be conducted so that an interpretation of all included studies can be presented.

Two reviewers will lead data synthesis. Emergent analysis, and any discrepancies, will be discussed with other members of the review team. Microsoft Office software will be used to facilitate data synthesis.

## Patient and public involvement

No patients or members of the public were involved in the design of this study.

## Amendments

An amendment has been made to the initial registration of this systematic review in PROSPERO, which originally stated that studies from both HICs and LMICs would be included in the review. The PROSPERO record was amended to state that only studies from LMICs will be included in this review, and studies from HICs will be excluded from this review. Any further amendments to this protocol will be documented in the full review.

## ETHICS AND DISSEMINATION

Ethics approval is not required as this is a protocol for the systematic review of previously published data. In addition to a report to the funding body, we intend to submit the systematic review for publication in a high-impact, peer-reviewed journal. We will select an open access journal to ensure free access to undergraduate and graduate students, researchers, academics and research groups. Results will also be disseminated at conferences seminars, congresses and symposia, and to relevant stakeholders.

**Acknowledgements** The authors would like to thank Mrs Nia Morris from Betsi Cadwaladr University Health Board library for her contribution to the development of the search strategy.

**Contributors** CR, RM, PH, AK and MK conceived the research idea. SN led the development of the search strategy, with input from Nia Morris, RM, AK and PH. RM, SN, MK, AK, PH and CR conceived the manuscript. RM led the drafting of the manuscript with contributions from SN, AE, SB, MK, AK, PH, NC and CR. All authors approved the final manuscript.

**Funding** This manuscript was funded by UK Research and Innovation Global Challenge Research Fund (ref: MR/P028144/1). The funder did not have any role in developing this protocol.

**Competing interests** None declared.

**Patient consent for publication** Not required.

**Provenance and peer review** Not commissioned; externally peer-reviewed.

**ORCID iDs**
Rebecca McPhillips http://orcid.org/0000-0003-4296-5970
Sadia Nafees http://orcid.org/0000-0003-1553-3013
Anne Krayer http://orcid.org/0000-0003-1503-1734

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
