## [Reviewer comments · BMJ Open]

ARTICLE DETAILS

TITLE (PROVISIONAL)	Knowledge, attitudes, and experiences of self-harm and suicide in low and middle income countries: protocol for a systematic review
AUTHORS	McPhillips, Rebecca; Nafees, Sadia; Elahi, Anam; Batool, Saqba; Krishna, Murali; Krayner, Anne; Huxley, Peter; Chaudhry, Nasim; Robinson, Catherine

VERSION 1 – REVIEW

REVIEWER	Visentin, Denis University of Tasmania, College of Health and Medicine
REVIEW RETURNED	09-Sep-2020

GENERAL COMMENTS	This study reports a protocol for a systematic review regarding stakeholder’s knowledge, attitudes, and experiences of self-harm and suicide in low and middle income countries. This is a topic worth of a review, and would inform policy in this area. However, the background to the research, and the scope of the study needs to be more clearly defined in the protocol. Strengths/Limitations The authors report a strength of the review as a “mixed-methods approach”. The review may assess both quantitative and qualitative evidence but this should not be described as mixed methods for a review. A strength identified was the development of a community survey towards self-harm and suicide in South Asia. This should be removed as it does not directly align with the LMIC focus of the review. Introduction Page 3 In 52. “...for each adult that dies from suicide there may be 20 more suicide attempts.” Can the authors please clarify if this refers to 20 other individuals or 20 other attempts from the same individual (or a combination of both). The authors have not adequately provided background information as to who the stakeholders are as distinct from the previous existing research on healthcare professionals. The definition of stakeholders is important for the review search and inclusion criteria. Given that there is already existing research regarding healthcare professionals and that healthcare professionals form part of the stakeholders
---

	in this review, the information about the stakeholders is important. The introduction focusses heavily on suicide with self-harm introduced only as a risk factor for suicide. As the review includes self-harm as distinct from suicide, it is important that self-harm is described in the introduction more fully, and in the context of stakeholders and LMICs. Methods and Analysis The stakeholder terms in the search do not appear to be exhaustive (the term stakeholder does not appear for example). Other terms do not cover stakeholders such as policy makers, government and NGOs, health service – which were stated in the inclusion criteria. Similarly, the knowledge and attitude search terms have some particularities that have not been described in the protocol. What types of knowledge and attitudes are the researchers interested in? The statement regarding the non-exclusion of studies is open to interpretation. While many reviews do not exclude studies based on poor quality ratings, the statement here seems to imply that a combination of the quality ratings and the “common sense” of the study authors will be used to determine inclusion. It would be preferable to include all studies that meet the criteria and to discuss the quality of the evidence arising in the review itself – or otherwise use the quality ratings to exclude studies. Otherwise the process is not transparent and open to reviewer bias (real or perceived). An explanation is required as to why the age of 16 is used as a cutoff. This may exclude many studies which consider youth suicide for which the data from those over the age of 16 cannot be separately be extracted. The definition of “youth” can vary widely across studies and may. For example see the review of Grimmond J, et al (2019) A qualitative systematic review of experiences and perceptions of youth suicide. PLoS ONE 14(6): e0217568. This shows the range of definitions of youth which may make extraction difficult for a review protocol with a cutoff of 16 years, which would be similar for quantitative studies. Also the interpretation of the cutoff of 16 may be problematic if the purpose of the review is to consider adult suicide and self harm. There are many differences in the knowledge and attitudes towards youth suicide as distinct from adult suicide. The authors would need to justify this in the study protocol.
--	---

REVIEWER	Colucci, Erminia Middlesex University
REVIEW RETURNED	18-Nov-2020

GENERAL COMMENTS	This is an excellent and sound protocol that focuses on an important topic and population. I strongly recommend it for
--

	publication after making or considering the following amendments: 1- please do not use the work commit or committing suicide; 2-stakeholders should also include peer, psych*, lived experience 3-LMIC should include South Asia
--	--

VERSION 1 – AUTHOR RESPONSE

Reviewer comment: This study reports a protocol for a systematic review regarding stakeholder’s knowledge, attitudes, and experiences of self-harm and suicide in low and middle income countries. This is a topic worth of a review, and would inform policy in this area. However, the background to the research, and the scope of the study needs to be more clearly defined in the protocol.

Response: The background to the research and the scope of the study have been more clearly defined in the manuscript (pages 3-5).

Reviewer comment: The authors report a strength of the review as a “mixed-methods approach”. The review may assess both quantitative and qualitative evidence but this should not be described as mixed methods for a review.

Response: The review is no longer described as ‘mixed methods’ (see deletions page 3, lines 7 & 16)

Reviewer comment: A strength identified was the development of a community survey towards self-harm and suicide in South Asia. This should be removed as it does not directly align with the LMIC focus of the review.

Response: The development of a community survey as a strength has been removed (see deletion page 3, lines 12-13)

Reviewer comment: Introduction Page 3 In 52. “...for each adult that dies from suicide there may be 20 more suicide attempts.” Can the authors please clarify if this refers to 20 other individuals or 20 other attempts from the same individual (or a combination of both).

Response: This refers to 20 other individuals. This has been clarified in the manuscript (page 4, line 11).

Reviewer comment: The authors have not adequately provided background information as to who the stakeholders are as distinct from the previous existing research on healthcare professionals. The definition of stakeholders is important for the review search and inclusion criteria. Given that there is already existing research regarding healthcare professionals and that healthcare professionals form part of the stakeholders in this review, the information about the stakeholders is important.

Response: The definition of stakeholders has now been included in the introduction

(page 5, lines 19-21), and is also included in the study selection criteria section (page 7, lines 7-10).

Reviewers comment: The introduction focusses heavily on suicide with self-harm introduced only as a risk factor for suicide. As the review includes self-harm as distinct from suicide, it is important that self-harm is described in the introduction more fully, and in the context of stakeholders and LMICs.

Response: Self-harm has been described more fully in the introduction, and in the context of LMICs (pages 4, lines 13-27).

Reviewers comment: The stakeholder terms in the search do not appear to be exhaustive (the term stakeholder does not appear for example). Other terms do not cover stakeholders such as policy makers, government and NGOs, health service – which were stated in the inclusion criteria.

Response: The search strategy was developed with the assistance of a librarian from Betsi Cadwaladr University Health Board library. The terms communit\$, societ\$, government\$, health personnel, physicians, personnel hospital are intended to capture stakeholders such as policy makers, government and NGOs and those in the health services.

Reviewers comment: Similarly, the knowledge and attitude search terms have some particularities that have not been described in the protocol. What types of knowledge and attitudes are the researchers interested in?

Response: We are interested in a broad spectrum of knowledge, attitudes and experiences that various stakeholders may have concerning self-harm. This is now stated in the protocol, and examples are provided (pages 5, lines 18-26).

Reviewers comment: The statement regarding the non-exclusion of studies is open to interpretation. While many reviews do not exclude studies based on poor quality ratings, the statement here seems to imply that a combination of the quality ratings and the “common sense” of the study authors will be used to determine inclusion. It would be preferable to include all studies that meet the criteria and to discuss the quality of the evidence arising in the review itself – or otherwise use the quality ratings to exclude studies. Otherwise the process is not transparent and open to reviewer bias (real or perceived).

Response: We agree that the previous statement regarding non-exclusion of studies was open to interpretation. Our intention is to include all studies as the reviewer suggests and discuss the quality of evidence in the review. This has been clarified in the protocol (page 9, lines 16-21).

Reviewers comment: An explanation is required as to why the age of 16 is used as a cutoff. This may exclude many studies which consider youth suicide for which the data from those over the age of 16 cannot be separately be extracted. The definition of “youth” can vary widely across studies and may. For example see the review of Grimmond J, et al (2019) A qualitative systematic review of experiences and perceptions of youth suicide. PLoS ONE 14(6): e0217568. This shows the range of definitions of

youth which may make extraction difficult for a review protocol with a cutoff of 16 years, which would be similar for quantitative studies. Also the interpretation of the cutoff of 16 may be problematic if the purpose of the review is to consider adult suicide and self harm. There are many differences in the knowledge and attitudes towards youth suicide as distinct from adult suicide. The authors would need to justify this in the study protocol.

Response: We agree that the definition of youth can vary across studies and indeed cultures. The cut off point of 16 is for stakeholders who have participated in research, as 16 is widely regarded as the age that individuals without impairment can consent to be involved with research (see British Psychology Society: <https://www.bps.org.uk/sites/www.bps.org.uk/files/Policy/Policy%20-%20Files/BPS%20Code%20of%20Human%20Research%20Ethics.pdf>). Research that concerns stakeholders' (aged 16 and over) knowledge, attitudes and experiences of suicide and self-harm related to those who are not aged 16 and above will be included in the review if all inclusion criteria are met.

We recognise that there may be studies where data from those over the age of 16 cannot be extracted and if this is the case we will reflect on this limitation in our paper.

We have included further information on this cut off point in the protocol (page 7, lines 6-7).

Reviewers comment: Please do not use the work commit or committing suicide

Response: The word 'committed' has been replaced with 'completed' (page 6, line 9)

Reviewers comment: Stakeholders should also include peer, psych*, lived experience

Response: Preliminary searches have been performed using the search strategy detailed in Appendix 1 (see Prospero ID: CRD42019135323) and studies on lived experience and peer experiences have been captured.

Reviewers comment: LMIC should include South Asia

Response: The term 'Asia' is the parent term in the MeSH heading tree and captures studies conducted in South Asia.

VERSION 2 – REVIEW

REVIEWER	Visentin, Denis University of Tasmania, College of Health and Medicine
REVIEW RETURNED	11-Jan-2021
GENERAL COMMENTS	BMJ Reviewers Response - Knowledge, attitudes, and experiences of self-harm and suicide in low and middle income countries: protocol for a systematic review I thank the authors for considering carefully the comments and concerns raised. Where the authors have agreed with

	the comments, they have made changes that have addressed these sufficiently. Where we have differed in opinion, they have provided considered responses, and have included enough information in the manuscript for the reader to understand their approach. I still have some concerns regarding the 16 years of age cutoff, and this issue will need to be carefully considered in the data which is extracted from each study. As the authors note, there are likely to be studies which include participants above and below the cutoff, but this would be the case whichever cutoff is chosen. A few minor considerations in the revised manuscript Page 3 Line 9. Suggest change "We anticipate..." to As it is likely that..." Page 4 Line 3 "...where marriage..." should be "...while marriage..." Page 4 Par 1. This section is a little confusing. It introduces NSSI as distinct from "suicidal behaviours", however it also uses the term "self-harm". I think that self-harm should be defined clearly, as in many studies it is used to include harm arising from a suicidal behaviour. Hence the later comment that "self-harm has been found to be a robust predictor of suicidal behaviour" can be confusing. Also consider changing "While the intent of suicidal behaviours is to kill oneself..." I suggest "end one's life" or "take one's life" is better, and also you may consider taking a more in-depth approach to this by including that it is often the removal of suffering/pain by ending one's life as this is generally the prime motivator.
--	---

REVIEWER	Colucci, Erminia Middlesex University
REVIEW RETURNED	23-Feb-2021

GENERAL COMMENTS	Thank you for addressing mine and the other reviewers'comment. I believe this is a good quality and useful piece of work and recommend it for publication. All the best, I look forward to see your full review for which I volunteer to be a reviewer too!
---

VERSION 2 – AUTHOR RESPONSE

Reviewers comments	Response
I thank the authors for considering carefully the comments and concerns raised. Where the authors have agreed with the comments, they have made changes that have addressed these sufficiently. Where	We thank the reviewer for their feedback on age cut-off. This issue will be carefully considered during data extraction, analysis and when

we have differed in opinion, they have provided considered responses, and have included enough information in the manuscript for the reader to understand their approach. I still have some concerns regarding the 16 years of age cutoff, and this issue will need to be carefully considered in the data which is extracted from each study. As the authors note, there are likely to be studies which include participants above and below the cutoff, but this would be the case whichever cutoff is chosen.	disseminating the results of the review.
Page 3 Line 9. Suggest change “We anticipate...” to “As it is likely that...”	The beginning of the sentence on page 3, line 9, has been changed.
Page 4 Line 3 “...where marriage...” should be “...while marriage...”	The phrase has been changed to ‘while marriage’, on page 4, line 3.
Page 4 Par 1. This section is a little confusing. It introduces NSSI as distinct from “suicidal behaviours”, however it also uses the term “self-harm”. I think that self-harm should be defined clearly, as in many studies it is used to include harm arising from a suicidal behaviour. Hence the later comment that “self-harm has been found to be a robust predictor of suicidal behaviour” can be confusing. Also consider changing “While the intent of suicidal behaviours is to kill oneself...” I suggest “end one’s life” or “take one’s life” is better, and also you may consider taking a more in-depth approach to this by including that it is often the removal of suffering/pain by ending one’s life as this is generally the prime motivator.	We thank the reviewer for this feedback and upon re-reading agree that the use of the term ‘self-harm’ is confusing. We have made changes to this paragraph, page 4, lines 9-10, 16, 17, 19, 22, 23 in order to clarify the points made. The phrase has been changed to ‘end one’s life’ on page 4, line 13. We have also included that the motivation is ‘often to remove suffering’ on page 4, line 12.

In addition to the above outlined changes, we have also clarified the amendment that was made to the PROSPERO record, page 10, lines 11-12.

VERSION 3 – REVIEW

REVIEWER	Visentin, Denis University of Tasmania, College of Health and Medicine
REVIEW RETURNED	11-Mar-2021
GENERAL COMMENTS	I thank the authors for considering the suggestions in my previous review. These have been appropriately addressed and I have no further suggestions for improvement.